# Discovery of a maximally charged Weyl point

Qiaolu Chen[1,2,3], Fujia Chen[1,2,3], Yuang Pan[1,2,3], Chaoxi Cui[4,5], Qinghui Yan[1,2,3], Li Zhang[1,2,3], Zhen Gao[6], Shengyuan A. Yang [7], Zhi-Ming Yu [4,5] ✉, Hongsheng Chen [1,2,3] ✉, Baile Zhang [8,9] ✉ & Yihao Yang [1,2,3] ✉

The hypothetical Weyl particles in high-energy physics have been discovered in three-dimensional crystals as collective quasiparticle excitations near two-fold degenerate Weyl points. Such momentum-space Weyl particles carry quantised chiral charges, which can be measured by counting the number of Fermi arcs emanating from the corresponding Weyl points. It is known that merging unit-charged Weyl particles can create new ones with more charges. However, only very recently has it been realised that there is an upper limit — the maximal charge number that a two-fold Weyl point can host is four — achievable only in crystals without spin-orbit coupling. Here, we report the experimental realisation of such a maximally charged Weyl point in a three-dimensional photonic crystal. The four charges support quadruple-helicoid Fermi arcs, forming an unprecedented topology of two non-contractible loops in the surface Brillouin zone. The helicoid Fermi arcs also exhibit the long-pursued type-II van Hove singularities that can reside at arbitrary momenta. This discovery reveals a type of maximally charged Weyl particles beyond conventional topological particles in crystals.

There is a longstanding paradigm in condensed matter and other material fields that the low-energy excitations near certain band degeneracy points represent a platform for the emulation of exotic particles originally predicted in high-energy physics. A recent example is the Weyl points (WPs) (Fig. 1a, left panel) in three-dimensional (3D) band structures that represent the discovery of Weyl particles in a crystal environment, although their counterparts in high-energy physics remain hypothetical[1–5]. Such Weyl particles in crystals carry quantised chiral charges (Fig. 1b, left panel), as measured by the Chern number ($C$), and exhibit topologically protected helicoidal surface states (Fig. 1c, left panel), which form Fermi arcs at the Fermi level in

the surface Brillouin zone (Fig. 1d, left panel)[5]. Generally, Weyl particles carry a unit chiral charge ($|C| = 1$).

Moving beyond the framework of high-energy physics, the crystals' rich symmetry groups have helped to generalise Weyl particles into a big family of topological particles in crystals. While conventional Weyl particles have a spin of 1/2, in accord with the two-fold degeneracy of a Weyl point, unconventional topological particles can emerge near multifold band degeneracies in 3D crystals, behaving like particles with high spin[6–11]. For example, the recently discovered spin-1 topological particles are formed near a three-fold band degeneracy, exhibiting a chiral charge of two ($|C| = 2$)[7–11]. More recent studies have

[1]Interdisciplinary Centre for Quantum Information, State Key Laboratory of Modern Optical Instrumentation, ZJU-Hangzhou Global Scientific and Technological Innovation Centre, Zhejiang University, Hangzhou 310027, China. [2]International Joint Innovation Centre, Key Lab. of Advanced Micro/Nano Electronic Devices & Smart Systems of Zhejiang, The Electromagnetics Academy at Zhejiang University, Zhejiang University, Haining 314400, China. [3]Jinhua Institute of Zhejiang University, Zhejiang University, Jinhua 321099, China. [4]Centre for Quantum Physics, Key Laboratory of Advanced Optoelectronic Quantum Architecture and Measurement (MOE), School of Physics, Beijing Institute of Technology, Beijing 100081, China. [5]Beijing Key Laboratory of Nanophotonics and Ultrafine Optoelectronic Systems, School of Physics, Beijing Institute of Technology, Beijing 100081, China. [6]Department of Electrical and Electronic Engineering, Southern University of Science and Technology, Shenzhen 518055, China. [7]Research Laboratory for Quantum Materials, Singapore University of Technology and Design, Singapore 487372, Singapore. [8]Division of Physics and Applied Physics, School of Physical and Mathematical Sciences, Nanyang Technological University, 21 Nanyang Link, Singapore 637371, Singapore. [9]Centre for Disruptive Photonic Technologies, The Photonics Institute, Nanyang Technological University, 50 Nanyang Avenue, Singapore 639798, Singapore. ✉e-mail: zhiming_yu@bit.edu.cn; hansomchen@zju.edu.cn; blzhang@ntu.edu.sg; yangyihao@zju.edu.cn

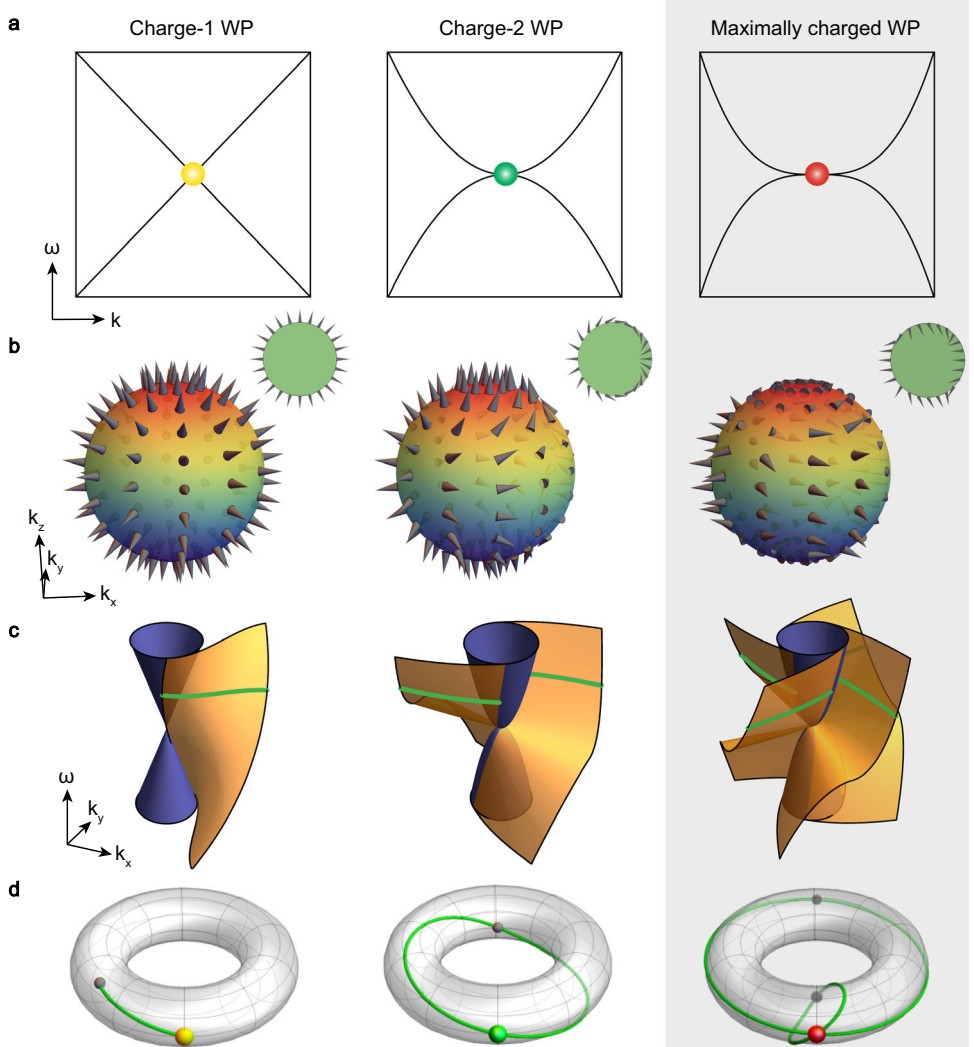

**Fig. 1 | Weyl points (WPs) with different topological charges. a** Schematic energy dispersions of the charge-1 WP (yellow dot), the charge-2 WP (green dot), and the maximally charged WP (red dot), respectively. **b** Spin structures of the charge-1 WP, the charge-2 WP, and the maximally charged WP, respectively. Insets are the top views of the spin structures (see Supplementary Note 1 for more details). **c** Surface dispersions near the projections of the charge-1 WP, the charge-2 WP, and the maximally charged WP, respectively. The solid blue cones, the helicoid sheets, and the green lines represent the projected bulk dispersions, the surface dispersions, and the Fermi arcs, respectively. The surface dispersions are topologically equivalent to the Riemann surface of $\mathrm{Im}\left[\log\sqrt[|C|]{k^{|C|}}\right]$, with $k = k_x + ik_y$ and $|C|$ being

the magnitude of topological charge (see Supplementary Note 2 for more details). **d** Typical Fermi arcs (green curves) wrapping around the surface Brillouin zone torus. For the charge-1 WP (yellow dot), the single Fermi arc forms an open curve; For the charge-2 WP (green dot), the double Fermi arcs can form a single noncontractible loop, characterised by $\mathbf{W} = \{1,0\}$ or $\{0,1\}$; For the maximally charged WP (red dot), the quadruple Fermi arcs can form double noncontractible loops, characterised by $\mathbf{W} = \{1,1\}$. These three cases are topologically distinct from each other. In this work, we mainly focus on the maximally charged WP (highlighted as a shaded area).

confirmed that the maximal chiral charge that such high-spin topological particles can carry is capped at four ($|C| = 4$), as recently demonstrated for spin-3/2 topological particles near a four-fold band degeneracy in the crystal palladium gallium (PdGa) with strong spin-orbit coupling[12]. However, it is unclear whether the chiral charge of spin-1/2 Weyl particles is subject to such an upper limit.

The answer to this question is nontrivial because it is known that merging two or more unit-charged Weyl particles is able to generate new ones with more charges[13]. For example, a charge-2 Weyl particle formed in a crystal with $C_4$ or $C_3$ symmetry can be treated as two charge-1 Weyl particles merged together[13–16]. The resultant dispersion of the two-fold degeneracy changes from linear to quadratic in the plane perpendicular to the $C_4$ or $C_3$ axis (Fig. 1a, middle panel). The two charges support double helicoids of surface states (Fig. 1c, middle panel), being able to form a non-contractible loop in the surface Brillouin zone at Fermi level

(Fig. 1d, middle panel). Charge-3 Weyl particles can be constructed similarly[13,14].

Only very recently has it been realised that there indeed exists an upper limit on the charge of Weyl particles[17–19]. After exhaustively examining all possible symmetry groups, it has been pointed out that a two-fold WP can maximally host a chiral charge of four[18,19], as previously missed in early theories of Weyl particles. The four charges exhibit complex spin texture analogous to a previously-overlooked higher-order skyrmion with a topological skyrmion number $N_{sk} = 4$ in real space (Fig. 1b, right panel)[20]. This previously missed Weyl particle disperses cubically along certain directions (Fig. 1a, right panel), hosting quadruple helicoids of surface states (Fig. 1c, right panel). In particular, the Fermi arcs at the Fermi level can form an unprecedented topology of two—the maximum possible number—noncontractible loops in the surface Brillouin zone (Fig. 1d, right panel). Unlike the maximally charged spin-3/2 topological particles recently discovered

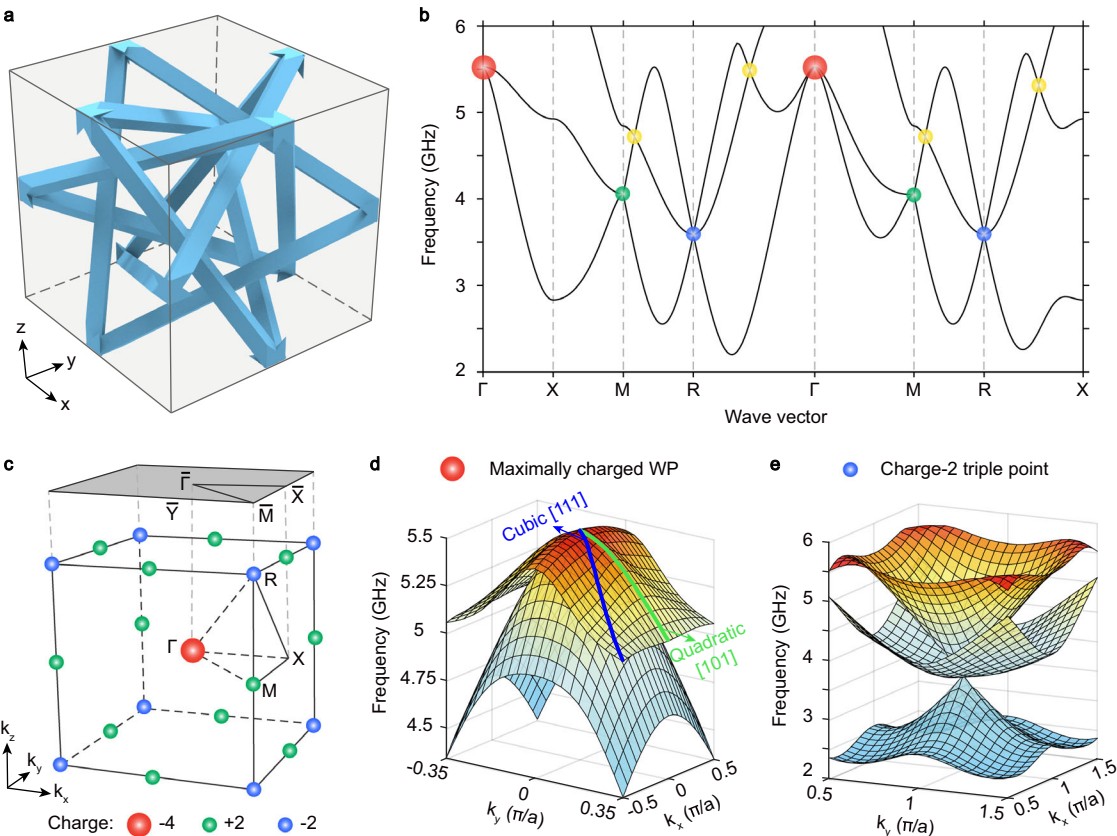

**Fig. 2 | 3D photonic crystal with the maximally charged Weyl point (WP).**
**a** A unit cell of the 3D metallic-mesh photonic crystal. **b, c** Band structure (**b**) and 3D first Brillouin zone (**c**) of the photonic crystal, with red, blue, green and yellow dots indicating the maximally charged WP, the charge-2 triple point, the charge-2 WPs, and the charge-1 WPs, respectively. **d** Two-dimensional (2D) band structure in the vicinity of the maximally charged WP, with cubic (quadratic) dispersion along [111] ([101]) direction. **e** 2D band structure in the vicinity of the charge-2 triple point.

in PdGa that rely on strong spin-orbit coupling[12], the maximally charged spin-1/2 Weyl particles can emerge only in crystals without spin-orbit coupling[17–19].

Here, we experimentally demonstrate such a charge-4 Weyl particle in a 3D photonic crystal. Apart from the maximally charged WP, our photonic crystal hosts a series of other topological nodal points, including multiple charge-1 WPs, multiple charge-2 WPs, and a charge-2 triple point—a three-fold degeneracy with chiral charge 2[21,22]. Via microwave pump-probe spectroscopies, we directly map out the projected bulk dispersion and the exotic quadruple-helicoid Fermi arcs of the topological surface states emanating from the maximally charged WP, verifying that the topological charge of this newly-discovered band degeneracy is indeed four. Remarkably, the quadruple Fermi arcs form double noncontractible loops wrapping around the surface Brillouin zone torus, characterised by two winding numbers $\mathbf{W} = \{w_x, w_y\} = \{1,1\}$, a previously-unidentified class of iso-frequency contours. Moreover, the helicoid arcs exhibit the long-pursued type-II van Hove singularities with saddle point dispersions that are not constrained to reside at time-reversal-invariant momenta[23].

## Design of a 3D photonic crystal with the maximally charged WP

The designed 3D photonic crystal has a cubic unit cell with lattice constant $a = 15$ mm, as shown in Fig. 2a. Each unit cell contains four equilateral metallic triangles with the vertices at the centre of twelve edges. Each vertex connects to eight adjacent vertices via square rods with length $l = \frac{\sqrt{6}}{2}a$ and width $w = 1$ mm. As it is a good approximation for metals to be treated as perfect electric conductors (PECs) at microwave frequencies, we set the equilateral triangles as PECs and the rest of the volume as air in numerical calculations. The resulting 3D photonic crystal has symmorphic space group P432 (No. 207) with $C_3$ rotation symmetry along [111] direction, and $C_4$ rotation symmetry along [100], [010], and [001] directions (see Supplementary Note 3 and Supplementary Fig. 12 for the detailed design). The space group symmetry of this photonic crystal can be generated by the following five symmetry operators: a three-fold rotation along [111] direction $C_{3,111}$, three two-fold rotations along [001] direction $C_{2z}$, [100] direction $C_{2x}$, and [110] direction $C_{2,110}$, and time-reversal symmetry $\tau$. Since the space group P432 does not have inversion symmetry, it can host chiral topological point degeneracies at time-reversal-invariant momenta.

Employing the first-principles method, we numerically calculate the band structures of the 3D photonic crystal, as shown in Fig. 2b–e. The first three bands cross with each other, forming a maximally charged WP at $\Gamma$ (red dots), three charge-2 WPs at M (green dots), a charge-2 triple point at R (blue dots), and multiple charge-1 WPs at general momenta (yellow dots). In this work, we mainly focus on the maximally charged WP that is protected by $C_{3,111}$, $C_4$ rotation symmetry and time-reversal symmetry $\tau$ of the underlying photonic crystal. The maximally charged WP features a cubical dispersion along the four three-fold rotation axes and a quadratic dispersion along other directions, as shown in Fig. 2b, d. Its 2 × 2 low-energy effective Hamiltonian around $\Gamma$ can be described by[19]

$$H_\Gamma = c_1 + c_2 k^2 + \begin{bmatrix} \frac{k_x^2 + k_y^2 - 2k_z^2}{\sqrt{3}}c_3 & c_3\left(k_x^2 - k_y^2\right) + ic_4 k_x k_y k_z \\ c_3\left(k_x^2 - k_y^2\right) - ic_4 k_x k_y k_z & -\frac{k_x^2 + k_y^2 - 2k_z^2}{\sqrt{3}}c_3 \end{bmatrix},$$

where $c_i$ ($i = 1, 2, 3, 4$) are real parameters, and $k = \sqrt{k_x^2 + k_y^2 + k_z^2}$ (see Supplementary Notes 4, 5 and 6 for the symmetry analysis and

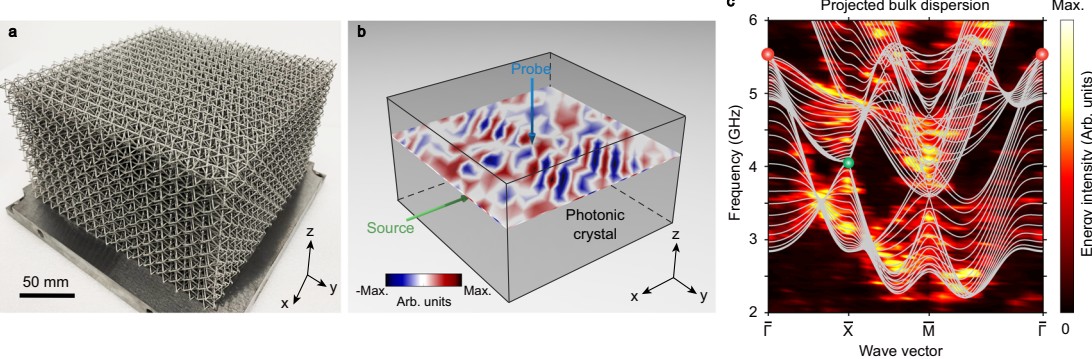

**Fig. 3 | Measurement of the projected bulk dispersion of the maximally charged Weyl point (WP). a** Photograph of the fabricated 3D photonic crystal, consisting of $17 \times 17 \times 10$ unit cells. **b** Experimental setup. The field pattern is measured on (001) surface of the sample. Green and blue arrows represent the source and probe, respectively. **c** Measured projected bulk dispersion along the high-symmetry line $\bar{\Gamma} - \bar{X} - \bar{M} - \bar{\Gamma}$. Grey curves display the numerically-calculated projected bulk dispersion. Red (green) dots represent the projections of the maximally charged WP (charge-2 WP). The colourmap measures the energy density.

effective Hamiltonian descriptions). By employing the Wilson loop method, we numerically confirm that the topological charge of the maximally charged WP is −4 (see Supplementary Note 7). Besides, our photonic crystal hosts the charge-2 triple point at R (see Fig. 2e). Its low-energy effective Hamiltonian denotes $H_R = c_5 + c_6 \, \mathbf{k} \cdot \mathbf{L}$ with $c_i$ ($i = 5$, 6) being real parameters and $\mathbf{L} = \{L_x, L_y, L_z\}$ being the spin-1 matrix representation (see Supplementary Note 4). The first-principles calculation manifests that the topological charges of the triple point at R and the charge-2 WPs at M are −2 and +2, respectively (see Supplementary Note 7). Therefore, for the first band, the total topological charge is zero, which is consistent with the no-go theorem[24].

## Experimental observation of the maximally charged WP and its quadruple-helicoid surface states

Due to the self-supporting architecture of our photonic crystal, the sample is directly fabricated via metallic additive manufacturing without relying on supporting structures. The material is stainless steel (316 L) with relatively high conductivity at microwave frequencies, and the sample has $17 \times 17 \times 10$ unit cells, as shown in Fig. 3a.

Next, we perform microwave pump-probe measurements to determine the projected bulk dispersion. As schematically shown in Fig. 3b, the point-like source is horizontally inserted into the fifth layer through the air holes of the 3D photonic crystal, to excite the bulk modes. We also vertically thread an electric dipole antenna into different layers to probe the field distributions on (001) surface. The amplitude and phase of the local fields at each frequency are collected by a vector network analyser (VNA). The field distributions are scanned point by point with a spatial resolution of 15 mm in both [100] and [010] directions (see Supplementary Fig. 14).

We focus on the projected bulk dispersion along the high-symmetry line $\bar{\Gamma} - \bar{X} - \bar{M} - \bar{\Gamma}$, which is obtained by projecting the 3D band structure onto the $k_x$-$k_y$ plane. Consequently, the maximally charged WP is projected onto the centre of the surface Brillouin zone (i.e., $\bar{\Gamma}$), while two charge-2 WPs are projected onto the edge centre of the surface Brillouin zone (e.g., $\bar{X}$). One charge-2 WP and the charge-2 triple point are projected onto the corner of the surface Brillouin zone (i.e., $\bar{M}$). By applying the 2D spatial Fourier transform to the real-space complex field distributions, we obtain the projected bulk dispersion, as shown in Fig. 3c. For reference, the numerically-calculated projected bulk dispersion is also plotted, which is marked by the grey curves in Fig. 3c. One can see that the experimental results roughly match the simulated counterparts.

To confirm the topological charge of the maximally charged WP, we use surface measurement to probe the field distributions on the sample's (001) surface. The experimental configuration is shown in Fig. 4a, where the source (probe) lies on the second (first) layer of the sample. After Fourier-transforming the field data from real space to reciprocal space, the resulting intensity map of the surface dispersion along high-symmetry line $\bar{\Gamma} - \bar{X} - \bar{M} - \bar{\Gamma}$ is directly compared with the numerical results, as shown in Fig. 4b. It is evident that the topological surface states residing in bandgaps are consistently seen in both simulated (green curves) and experimental data from 3.7 to 4.5 GHz. Note that, apart from the topological surface states, a trivial surface mode exists at low frequencies (blue curve).

The topological charges of the maximally charged WP and the charge-2 WPs can be verified from the measured surface iso-frequency contours at the frequencies ranging from 3.7 to 4.5 GHz (see Fig. 4d–m). One can observe the quadruple (double) helicoid surface sheets of the topological surface states wind around $\bar{\Gamma}$ ($\bar{X}$ or $\bar{Y}$). It is obvious from the measured results that, at a fixed frequency (e.g., 4.1 GHz), four Fermi arcs connect $\bar{\Gamma}$ and $\bar{X}$ or $\bar{Y}$, proving that the topological charge of the maximally charged WP projected to $\bar{\Gamma}$ is indeed 4. Besides, two Fermi arcs from $\bar{X}$ or $\bar{Y}$ indicate the topological charge of the charge-2 WP projected to $\bar{X}$ or $\bar{Y}$ is 2. The topological charge at $\bar{M}$ is 0, owing to the charge offset between the third charge-2 WP ($C = +2$) and the charge-2 triple point ($C = -2$). One can also see that the four Fermi arcs collapse at the centre of the Brillouin zone at 4.5 GHz, where the gaps between the bulk iso-frequency contours enclosing the maximally charged WP at $\Gamma$ and the charge-2 WPs at M are closed. Thus, no Fermi arc states are observed above 4.5 GHz. Note that, since the charge-1 WPs involve the second and the third bulk bands, they do not affect the profiles of the above iso-frequency contours of Fermi arc states that exist in the gaps between the projections of the first and the second bulk bands. Besides, the double Fermi arcs of the charge-2 triple point are in principle accessible by projecting the triple point onto the (101) surface (see Supplementary Note 9).

Remarkably, the quadruple Fermi arcs wrap around the torus of the surface Brillouin zone, forming double noncontractible loops (see the right panel of Fig. 1d). Compared to the generally short Fermi arcs in conventional charge-1 Weyl crystals, the unconventional double noncontractible loops exhibit long topological Fermi arcs spanning a large area of the surface Brillouin zone, especially from 4.0 to 4.3 GHz in our case (see Fig. 4h–k). Mathematically, closed loops on a 2D surface Brillouin zone torus $T^2 = S^1 \times S^1$ can be classified by its fundamental homotopy group $\pi_1(T^2) \cong \pi_1(S^1) \oplus \pi_1(S^1) = \mathbb{Z} \oplus \mathbb{Z}$, labeled by two integers that indicate the number of times the loop winds around two orthogonal directions of the surface Brillouin zone, respectively. Our quadruple-helicoid surface states from 3.7 to 4.5 GHz

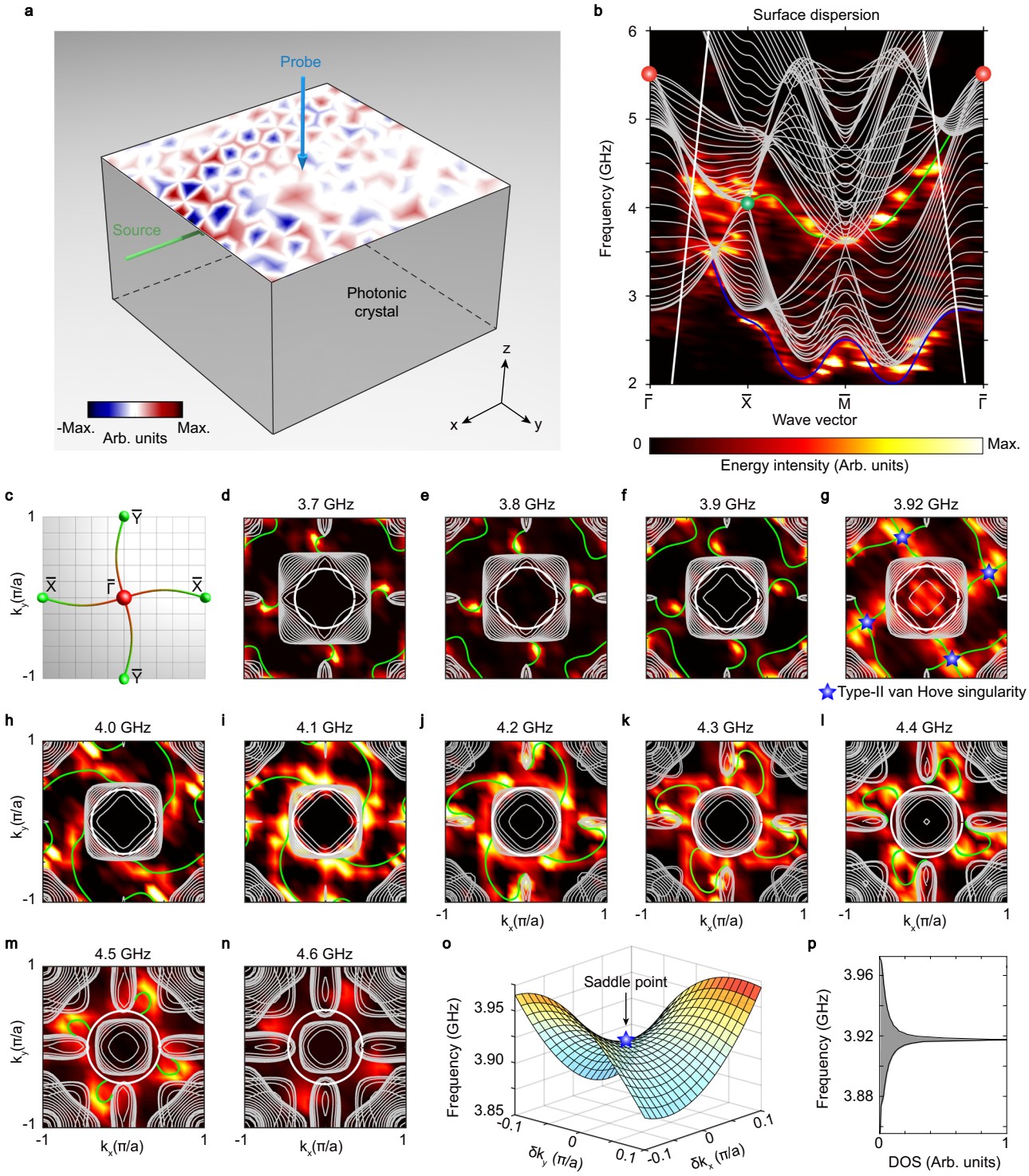

**Fig. 4 | Experimental observation of the quadruple-helicoid topological surface states of the maximally charged Weyl point (WP). a** Experimental setup. The field pattern is measured on the top (001) surface of the sample. Green and blue arrows represent the source and probe, respectively. **b** Measured surface dispersion along the high-symmetry line $\bar{\Gamma} - \bar{X} - \bar{M} - \bar{\Gamma}$. Green (blue) curves represent the topological surface states (trivial surface state). Grey curves display the numerically-calculated bulk states. Bold white curves show the light line. Red (green) dots display the projections of maximally charged WP (charge-2 WP). The colourmap measures the energy density. **c** Schematic of the quadruple Fermi arcs, connecting the projections of maximally charged WP (red dot at the centre) and charge-2 WPs (green dots at the edges). **d**–**n** Measured surface iso-frequency contours from 3.7 to 4.6 GHz. Green (grey) curves represent the topological surface states (bulk states). Bold white circles are the light cones. The colourmap measures the energy density. The plotted range for the iso-frequency contours is $[-\pi/a, \pi/a]^2$. **o** 2D band structure in the vicinity of the type-II van Hove singularity. Blue star indicates the saddle point. **p** Density of states (DOS) of the topological surface states.

are, thus, characterised by a nontrivial topological invariant of **W** = {$w_x, w_y$} = {1,1}. They are topologically distinct from the trivial surface states (whose iso-frequency contours can continuously deform

to a point, corresponding to a trivial class of {0,0}), the topological surface states of the conventional WPs (whose iso-frequency contours are open arcs, as shown in the left panel of Fig. 1d), and the topological

surface states of the previously-demonstrated unconventional nodal points (whose iso-frequency contours can be single noncontractible loops[9–11], corresponding to a nontrivial class of {1,0} or {0,1}, as shown in the middle panel of Fig. 1d).

The quadruple-helicoid surface states also exhibit type-II van Hove singularities in the density of states (DOS). One can see that the connectivity of the quadruple-helicoid arcs varies as the frequency increases (see Fig. 4d–n). Around 3.92 GHz, the helicoid arcs touch, forming saddle-like dispersions with the saddle points corresponding to the type-II van Hove singularities (see Fig. 4g, o). We also plot the DOS of the topological surface states in Fig. 4p, which shows a singularity at the frequency of the saddle point. Unlike their type-I counterparts constrained to reside at time-reversal-invariant momenta, the type-II van Hove singularities found here can locate at arbitrary momenta, which have not been demonstrated experimentally thus far[23,25]. Currently, the van Hove singularities are under active investigation in condensed-matter systems, such as twisted bilayer graphene[26,27], due to their electronic instabilities that lead to many unusual correlation effects (e.g., superconductivity). In particular, the type-II van Hove singularities are of great importance for realising topological odd superconductivity[23]. In photonics, the divergent DOS associated with the van Hove singularities can significantly enhance light-matter interaction, which is crucial for lasing, sensing, and quantum optics[28]. Therefore, we have demonstrated the long-pursued type-II van Hove singularities in a photonic topological medium.

## Discussion

We thus theoretically predict and experimentally discover the maximally charged WP with a chiral charge of four in a 3D photonic crystal, accompanied by a zoo of topological nodal points. The consequent quadruple-helicoid Fermi arcs emanating from the maximally charged WP are mapped out, which wrap around the surface Brillouin zone forming unprecedented double noncontractible loops. Our work establishes an ideal photonic platform for exploring novel physical phenomena related to the maximally charged WP, such as multiple chiral Landau levels[29], quantised circulation of the spatial shift in interface scattering[30], and unusual electromagnetic scattering near the WP[31]. Besides, our findings open the door to studying the potential applications of conventional and unconventional van Hove singularities, such as lasing and sensing, in the topological photonic media. When considering a super cell, the charge-2 triple point can be folded to Γ, providing a route to symmetry-enforced 3D metamaterials with vanishing refraction index[22]. Our work also offers a good opportunity to investigate exotic phenomena and realistic applications arising from the interplay between different types of topological nodal points in a single platform. Finally, our work will inspire research on the maximally charged topological band degeneracies in other physical settings, such as phononics, magnonics, and spinless electronics (see an acoustic design of the maximally charged WP in Supplementary Note 11).

## Methods

### Numerical simulation

We use the finite-element method solver of COMSOL Multiphysics software to perform the simulations. To calculate the 3D bulk dispersion and projected bulk dispersion of the photonic crystal, periodic boundary conditions are applied in all three spatial directions of the unit cell. For the surface dispersion, we consider an air-super-cell-air cladding structure composed of 18 unit cells and two 240-mm-thickness air boxes along [001] direction. In this case, all directions of the super cell have periodic boundary conditions, as the topological surface states are highly confined on the surface. The material of the 3D photonic crystal is considered as the PEC, and the rest volume is air.

## Experiment

Our experimental sample consists of $17 \times 17 \times 10$ unit cells, which is fabricated via the metallic additive manufacturing technique. The material is stainless steel 316 L (022Cr17Ni12Mo2), with relatively high conductivity at microwave frequencies. In the measurements, the amplitude and phase of the local fields at each frequency are collected by a VNA, as shown in Supplementary Fig. 14. The VNA is connected to two electric dipole antennas, serving as the source and the probe. For the bulk dispersion measurement, the source is horizontally inserted into the middle plane (the fifth layer) through the air holes of the 3D photonic crystal. The projected bulk dispersion shown in Fig. 3c can be regarded as the overlap of bulk dispersions on $k_x$-$k_y$ plane with different $k_z$. To obtain more bulk dispersions with different $k_z$, we vertically thread the probe into the second, third and fourth layers, respectively, and the projected bulk dispersion in Fig. 3c is averaged over the measured bulk dispersions on three layers. For the surface dispersion measurement, the source lies on the second layer of the photonic crystal to excite the surface modes, and we place the probe on the top layer. Note that, according to the symmetry of the 3D photonic crystal, (001), (100), and (010) planes are identical and support the same quadruple-helicoid surface states. We here adopt (001) plane in surface measurements. The field distributions are scanned point by point with a step of 15 mm along both [100] and [010] directions.

## Data availability

The data that support the findings of this study are available at https://doi.org/10.5281/zenodo.7299407.

## Code availability

The codes that support the findings of this study are available at https://doi.org/10.5281/zenodo.7299407.

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

## Acknowledgements

The work at Zhejiang University was sponsored by the Key Research and Development Program of the Ministry of Science and Technology under Grants No. 2022YFA1404704 (H.C.), 2022YFA1404902 (Y.Y.), and 2022YFA1405201 (Y.Y.), the National Natural Science Foundation of China (NNSFC) under Grants No.11961141010 (H.C.), No. 62175215 (Y.Y.), and No. 61975176 (H.C.), the Fundamental Research Funds for the Central Universities (2021FZZX001-19) (Y.Y.), and the Excellent Young Scientists Fund Program (Overseas) of China (Y.Y.). This work at Beijing Institute of Technology was sponsored by the National Natural Science Foundation of China (NNSFC) under Grants No. 12004035 (Z.Y.). The work at Nanyang Technological University was sponsored by the Singapore National Research Foundation (NRF) Competitive Research Program under Grant No. NRF-CRP23-2019-0007 (B.Z.), and Singapore Ministry of Education (MOE) Academic Research Fund Tier 3 under Grant No. MOE2016-T3-1-006 (B.Z.) and Tier 2 under Grant No. MOE2019-T2-2-085 (B.Z.).

## Author contributions

Y.Y., Z.Y. and B.Z. created the design. Q.C. and Y.Y. designed the experiment. Q.C. and Z.G. fabricated samples. Q.C. carried out the measurement with the assistance from F.C. and Y.P. Q.C. analysed the data. Q.C. and Y.Y. performed simulations with the input from L.Z. and S.Y. Q.C., Y.Y., H.C., Z.Y., C.C. and Q.Y. provided the theoretical explanations. Q.C. and Y.Y. wrote the manuscript with the input from B.Z., Z.Y., H.C. Y.Y., B.Z., H.C., and Z.Y. supervised the project. All authors contributed extensively to this work.

## Competing interests

The authors declare no competing interests.
