## [Peer Review File · Nature Communications]

Discovery of a maximally charged Weyl pointThis manuscript has been previously reviewed at another journal that is not operating a transparent peer review scheme. This document only contains reviewer comments and rebuttal letters for versions considered at *Nature Communications*.

REVIEWER COMMENTS

Reviewer #1 (Remarks to the Author):

The authors have provided very thorough and constructive replies to the points raised in the initial review round, including several new supplementary sections and supplementary figures, in addition to revised phrasing in relevant parts of the main text.

The manuscript - already excellent and reporting new, interesting, and important results - is even better and clearer as a consequence. I have thoroughly enjoyed reading the manuscript.

With this in mind, I fully support the publication of this manuscript in Nature Communications.

Reviewer #2 (Remarks to the Author):

In their reply, the authors have partly responded to my comments/questions in my previous report. I think this manuscript may be accepted for publication in Nature Communications, but with the following issues well addressed.

(1) In the comment 1 of my previous review report, I suggest the authors to give the detail of the $k \cdot p$ method for the effective Hamiltonian, but they only discuss it by giving some related references in the reply. I still strongly insist that the authors should give the detail, including the expansion based on the eigenstates and mode analysis, for the readers' benefit.

(2) In reply to comment 4, the authors write that, "The probe is vertically threaded into the second, third and fourth layers to measure more bulk states. The bulk dispersion in Fig. 3c is averaged over the experimental results on three layers." That are very confusing to me. Is it reasonable to obtain the bulk dispersion in such a way? The detailed descriptions for the measurement should be provided.

(3) In their reply to comment 6, the authors argue that the effect of all the charge-1 Weyl points can be ignored because they locate at different bands. But it seems to me that the Weyl points in the near bands can interact with one another and form Fermi arcs on surfaces, even they are not at the same band. The authors should label all the Weyl points in the related figures, and add some discussions about this important point. Furthermore, in Fig. R4, Fermi arcs disappear above 4.6 GHz, whereas the charge-4 Weyl point locates at 5.5 GHz, much higher than 4.6GHz, then how can the authors assert that the Fermi arcs are related to the charge-4 Weyl point?

Reviewer #3 (Remarks to the Author):

The authors have addressed all comments in detail. The additions to the manuscript, most notably the discussion on the tight-binding model of Note 3, now clearly elucidate the thought process followed for the design of the crystal and showcase its generality beyond the EM realm. Moreover, the addition of Note 9 provides a new intuitive viewpoint on how such k -space nodes can evolve and, by extension, manifest themselves in this platform. Considering that all technical details have also been addressed I recommend publication in Nature Communications.

Response Letter to Reviewers

We really appreciate the valuable comments given by all the reviewers on this manuscript (NCOMMS-22-35090-T).

In the following text, we quote each comment from each reviewer in *italics* and reply one by one. Accordingly, we also revised the main manuscript and Supporting Information while. These updates are highlighted in **blue**. In the text below, the references to these updates are highlighted in a similar way (i.e., highlighted in **blue**).

GENERAL COMMENTS FROM 1st REVIEWER:

The authors have provided very thorough and constructive replies to the points raised in the initial review round, including several new supplementary sections and supplementary figures, in addition to revised phrasing in relevant parts of the main text.

The manuscript - already excellent and reporting new, interesting, and important results - is even better and clearer as a consequence. I have thoroughly enjoyed reading the manuscript.

With this in mind, I fully support the publication of this manuscript in Nature Communications.

Response from Authors:

We thank the reviewer for the recommendation for the publication of this work.

GENERAL COMMENTS FROM 2nd REVIEWER:

In their reply, the authors have partly responded to my comments/questions in my previous report. I think this manuscript may be accepted for publication in Nature Communications, but with the following issues well addressed.

Response from Authors:

We thank the reviewer for the recommendation for the publication of this work. In the following, we fully address the specific comments point-by-point.

SPECIFIC COMMENTS FROM 2nd REVIEWER:

2nd Reviewer -- Comment 1:

In the comment 1 of my previous review report, I suggest the authors to give the detail of the $k.p$ method for the effective Hamiltonian, but they only discuss it by giving some related references in the reply. I still strongly insist that the authors should give the detail, including the expansion based on the eigenstates and mode analysis, for the readers' benefit.

Response from Authors:

Following the reviewer's suggestion, we give the details for the derivation of low-energy effective Hamiltonian at the maximally charged Weyl point as an example, including mode

analysis and numerical calculation of the matrix representations of the generators of the little group at Γ point.

We calculate the matrix representations (Eq. 18-21 in Supplementary Note 4) by utilizing first-principles simulated electric fields of two degenerate eigenmodes at the maximally charged Weyl point in the commercial software COMSOL, i.e., $\vec{E}_{01}(r)$ and $\vec{E}_{02}(r)$ (see Fig. R1). The normalized electric fields of the modes are,

$$\vec{E}_i(r) = \frac{\vec{E}_{0i}(r)}{\left(\int dr \vec{E}_{0i}^*(r) \vec{E}_{0i}(r) dr\right)^{\frac{1}{2}}} \quad (i = 1,2). \quad (1)$$

Fig. R1 | Simulated electric and magnetic fields of two degenerate modes at the maximally charged Weyl point.

Since the two modes are degenerate, an arbitrary linear combination of these two modes is also one of the eigenmodes of the degenerate point. In order to obtain the same matrix representations as that in Supplementary Note 4 (Eq. 18-21), we apply a unitary transformation on $\vec{E}_i(r)$ to $\vec{E}'_i(r)$. The unitary transformation could be written as,

$$\begin{pmatrix} \vec{E}'_1(r) \\ \vec{E}'_2(r) \end{pmatrix} = U \begin{pmatrix} \vec{E}_1(r) \\ \vec{E}_2(r) \end{pmatrix}. \quad (2)$$

Based on $\vec{E}'_i(r)$, the matrix representations $D(R)$ corresponding to the symmetric operation R can be derived as,

$$D(R) = \begin{pmatrix} \int \vec{E}'_1(r) \overline{R\vec{E}'_1}(R^{-1}r) dr & \int \vec{E}'_1(r) \overline{R\vec{E}'_2}(R^{-1}r) dr \\ \int \vec{E}'_2(r) \overline{R\vec{E}'_1}(R^{-1}r) dr & \int \vec{E}'_2(r) \overline{R\vec{E}'_2}(R^{-1}r) dr \end{pmatrix} \quad (3)$$

where $\overrightarrow{RE'_i}$ represents the operation that applies symmetric operation R to \vec{E}'_i , and $R^{-1}r$ is to apply inversed symmetric operation R^{-1} to r . Thus, we numerically obtain the four matrix representations for the maximally charged Weyl point,

$$C_{3,111} \approx \frac{1}{2} \begin{pmatrix} -1 & \sqrt{3} \\ -\sqrt{3} & -1 \end{pmatrix}, \quad (4)$$

$$C_{2z} \approx \begin{pmatrix} 1 & 0 \\ 0 & 1 \end{pmatrix}, \quad (5)$$

$$C_{2x} \approx \begin{pmatrix} 1 & 0 \\ 0 & 1 \end{pmatrix}, \quad (6)$$

$$C_{2,110} \approx \begin{pmatrix} 1 & 0 \\ 0 & -1 \end{pmatrix}. \quad (7)$$

One can see that the matrix representations are consistent with the theoretical matrix representations Eq. 18-21 in Supplementary Note 4.

In summary, based on the numerically simulated electric fields of the modes at the maximally charged Weyl point, we calculate the corresponding matrix representations of the generators of the little group, which are the same as the theoretical analysis (i.e., Eq. 18-21 in Supplementary Note 4). We thus can directly obtain the low-energy effective Hamiltonian (see the details in Supplementary Note 4). We note that the matrix representations for the charge-2 Weyl point and charge-2 triple point can also be derived in the same way.

Accordingly, we have added the above analysis and figures in the revised Supplementary Information as Supplementary Note 5 and Supplementary Fig. 2, on pages 13-14.

2nd Reviewer -- Comment 2:

In reply to comment 4, the authors write that, "The probe is vertically threaded into the second, third and fourth layers to measure more bulk states. The bulk dispersion in Fig. 3c is averaged over the experimental results on three layers." That are very confusing to me. Is it reasonable to obtain the bulk dispersion in such a way? The detailed descriptions for the measurement should be provided.

Response from Authors:

The projected bulk dispersion shown in Fig. 3c can be regarded as the overlap of bulk dispersion on k_x - k_y plane with different k_z . Therefore, by measuring the field distributions on different layers (along z direction), we can obtain the bulk dispersions with more k_z . In Fig. R2, we plot the measured bulk dispersions on the second, third and fourth layers, and the averaged bulk dispersion, clearly showing that the averaged bulk dispersion contains more bulk states.

Following the reviewer's suggestion, we have added detailed descriptions to make it clear in the main text, starting from line 325, page 13, which reads as,

"The projected bulk dispersion shown in Fig. 3c can be regarded as the overlap of bulk dispersions on k_x - k_y plane with different k_z . To obtain more bulk dispersions with different k_z , we vertically thread the probe into the second, third and fourth layers, respectively, and the

projected bulk dispersion in Fig. 3c is averaged over the measured bulk dispersions on three layers.”

Fig. R2 | Measured projected bulk dispersions. The final projected bulk dispersion is averaged by the bulk dispersions on the second, third and fourth layers. Grey curves display the numerically-calculated projected bulk dispersion. The colourmap measures the energy density.

2nd Reviewer -- Comment 3:

In their reply to comment 6, the authors argue that the effect of all the charge-1 Weyl points can be ignored because they locate at different bands. But it seems to me that the Weyl points in the near bands can interact with one another and form Fermi arcs on surfaces, even they are not at the same band. The authors should label all the Weyl points in the related figures, and add some discussions about this important point. Furthermore, in Fig. R4, Fermi arcs disappear above 4.6 GHz, whereas the charge-4 Weyl point locates at 5.5 GHz, much higher than 4.6GHz, then how can the authors assert that the Fermi arcs are related to the charge-4 Weyl point?

Response from Authors:

Following the reviewer’s suggestion, we plot the projected Weyl points on the lowest three bands on k_x - k_y plane, as shown in Fig. R3a, where the red dot at $\bar{\Gamma}$ and green dots at \bar{X} involve the first and second bands, while other dots with dashed outlines involve the second and third bands. Fig. R3b displays the corresponding surface dispersion, which clearly indicates the Fermi arcs from the Weyl points between second and third bands are separated from that of the maximally charged Weyl points—the former exists roughly above 4.8 GHz, and the latter appears roughly below 4.8 GHz.

Fig. R3 | Projected WPs of the lowest three bands on k_x - k_y plane, and the corresponding surface states. **a**, Projected WPs on k_x - k_y plane. The red and green dots with solid outlines represent the WPs involving the first and second bands, while other dots with dashed outlines indicate the WPs involving the second and third bands. **b**, Surface dispersion along the high-symmetry line $\bar{\Gamma} - \bar{X} - \bar{M} - \bar{\Gamma}$. Green (blue) curves represent the topological surface states (trivial surface states). Grey curves display the bulk states.

In order to prove the Fermi arcs below 4.6 GHz emanate from the maximally charged Weyl point, we check the surface dispersion along a closed loop around $\bar{\Gamma}$ (depicted by a clockwise red dashed circle in Fig. R4a and Fig. R4c). Fig. R4b shows the surface dispersion with four topological surface states. Since the clockwise circular path around $\bar{\Gamma}$ has a radius $0.64 \pi/a$, inside which only the maximally charged Weyl point exists (see Fig. R4c); we can conclude that the four Fermi arcs indeed emanate from the maximally charged Weyl point.

Fig. R4 | Numerical evidence for the topological charges of the maximally charged WP. **a**, The 3D Brillouin zone and its surface projection to the k_x - k_y plane. The red dashed circle encircling the $\bar{\Gamma}$ point is the projection of the red tube oriented along k_z direction. **b**, The surface dispersions along the clockwise circular paths around the $\bar{\Gamma}$, with radii of $0.64 \pi/a$. The green curves represent the topological surface states. **c**, Surface iso-frequency contour at 4.2 GHz. Green (grey) curves represent the dispersions of topological surface states (bulk states). The red dashed circle checks the surface states along the clockwise circular path. The red and green dots with solid outlines represent the WPs involving the first and second bands, while other dots with dashed outlines indicate the WPs involving the second and third bands.

Following the reviewer's suggestion, we have added a discussion and Fig. R3 into the Supplementary Information (as Supplementary Note 12 and Supplementary Fig. 11) on page 28, which reads,

“Supplementary Note 12. The projections of WPs on the lowest three bands on k_x - k_y plane, and the surface dispersion.

In this section, we discuss the projected WPs on the lowest three bands and the corresponding surface states. Supplementary Fig. 11a displays the projected WPs on k_x - k_y plane, where the red dot at $\bar{\Gamma}$ and green dots at \bar{X} involve the first and second bands, while other dots with dashed outlines involve the second and third bands. Supplementary Fig. 11b plots the corresponding surface dispersion, which clearly indicates the Fermi arcs from the Weyl points between the second and third bands are separated from that of the maximally charged Weyl points—the former exists roughly above 4.8 GHz, and the latter appears roughly below 4.8 GHz.”

GENERAL COMMENTS FROM 3rd REVIEWER:

The authors have addressed all comments in detail. The additions to the manuscript, most notably the discussion on the tight-binding model of Note 3, now clearly elucidate the thought process followed for the design of the crystal and showcase its generality beyond the EM realm. Moreover, the addition of Note 9 provides a new intuitive viewpoint on how such k -space nodes can evolve and, by extension, manifest themselves in this platform. Considering that all technical details have also been addressed I recommend publication in Nature Communications.

Response from Authors:

We thank the reviewer for the recommendation for the publication of this work.

REVIEWERS' COMMENTS

Reviewer #2 (Remarks to the Author):

The authors of this manuscript have satisfactorily addressed my questions/comments, and now I would like to recommend the manuscript for publication.

Response Letter to Reviewers

We really appreciate the valuable comments given by all the reviewers on this manuscript (NCOMMS-22-35090A). In the following text, we quote each comment from each reviewer in *italics* and reply one by one.

GENERAL COMMENTS FROM 2nd REVIEWER:

The authors of this manuscript have satisfactorily addressed my questions/comments, and now I would like to recommend the manuscript for publication.

Response from Authors:

We thank the reviewer for the recommendation for the publication of this work.